# Improved Adaptive Multi-Objective Particle Swarm Optimization of Sensor Layout for Shape Sensing with Inverse Finite Element Method

**DOI:** 10.3390/s22145203

**Published:** 2022-07-12

**Authors:** Xiaohan Li, Shengtao Niu, Hong Bao, Naigang Hu

**Affiliations:** Key Laboratory of Electronic Equipment Structure Design, Ministry of Education, Xidian University, Xi’an 710071, China; xhanli@stu.xidian.edu.cn (X.L.); stniu@stu.xidian.edu.cn (S.N.); nghu@xidian.edu.cn (N.H.)

**Keywords:** multi-objective particle swarm optimization, external candidate solution set, inverse finite element method, deformation reconstruction, exploitation, exploration

## Abstract

The inverse finite element method (iFEM) is one of the most effective deformation reconstruction techniques for shape sensing, which is widely applied in structural health monitoring. The distribution of strain sensors affects the reconstruction accuracy of the structure in iFEM. This paper proposes a method to optimize the layout of sensors rationally. Firstly, this paper constructs a dual-objective model based on the accuracy and robustness indexes. Then, an improved adaptive multi-objective particle swarm optimization (IAMOPSO) algorithm is developed for this model, which introduces initialization strategy, the adaptive inertia weight strategy, the guided particle selection strategy and the external candidate solution (ECS) set maintenance strategy to multi-objective particle swarm optimization (MOPSO). Afterwards, the performance of IAMOPSO is verified by comparing with MOPSO applied on the existing inverse beam model. Finally, the IAMOPSO is employed to the deformation reconstruction of complex plate-beam model. The numerical and experimental results demonstrate that the IAMOPSO is an excellent tool for sensor layout in iFEM.

## 1. Introduction

The deformation of engineering structures under complex working conditions seriously affects the safety and life of the structure. In the past two decades, related scholars in the engineering field have begun to study structural health monitoring (SHM) technology [1,2,3,4]. The monitoring of the structure is divided into static monitoring and dynamic monitoring according to the state of the measured structure. Monitoring of static structures has lower requirements on real-time performance, which can be measured directly using measuring equipment, and focuses more on the analysis of the measurement data [5,6,7]. However, the monitoring of dynamic structures requires higher real-time performance. Structural deformation reconstruction is an important branch of dynamic structural health monitoring. In order to meet the requirements of practical engineering, the deformation reconstruction algorithm should be suitable under various boundary and working conditions.

In Gherlone’s analysis on shape sensing of structural deformation [8], the inverse finite element method (iFEM) is an effective deformation reconstruction method. Based on the least square theory, iFEM minimizes the error between surface theoretical strain and actual strain, and then realizes the conversion of surface strain and displacement, which was proposed by Tessler et al. [9,10]. Afterward, Gherlone et al. utilized the mathematical concepts of the iFEM methodology to developed an inverse frame element based on the kinematic assumptions of Timoshenko beam theory, which reconstructed the mode of stretching, bending, transverse-shear, and torsion-deformation [11]. Then, Gherlone et al. numerically and experimentally validated the superior capability of the inverse-frame element for shape sensing of three-dimensional frame structures undergoing static or dynamic loading [12]. Zhao et al. reconstructed the displacement fields without the knowledge of structural mode shapes, material properties, and applied loading [13]. Niu et al. reconstructed the structural displacements based on strain data collected from the single surface of the structure [14]. Moreover, Chen et al. proposed a unified reconstruction method for the beam-like structure based on the framework of the iFEM, which reduces the structure identification error by adopting some generalized quantities [15].

In the application of the iFEM, some scholars found that the distribution of strain sensors will affect the reconstruction effect. In order to reduce the influence of external environmental factors on the measurement strain, Yuen et al. proposed a Bayesian sequential sensor placement algorithm for multi-type of sensors, based on the robust information entropy [16]. Zhao et al. adopted the particle swarm optimization algorithm to optimize the sensor placement, but it may lead to the degradation of the reconstruction accuracy [17]. Afterwards, Zhao et al. introduced multi-objective particle swarm optimization (MOPSO) to optimize the layout of the sensor, which optimizes both the robustness and the accuracy simultaneously [18]. However, due to the lack of a strategy to jump out of the local optimum, MOPSO is prone to ineffective exploration, with poor effect and low efficiency. The goal of this paper is to propose an optimization algorithm that can be better applied to sensor layout problems.

Optimization of sensor layout in iFEM is a multi-objective optimization problem where model gradient information is not easy to obtain. Traditional methods such as the Conjugate gradient method are no longer applicable. However, with the development of computer technology and biological research, evolutionary algorithms (EA) are gradually used in complex optimization problems. EA is a kind of heuristic random search algorithm based on population evolution, which has the ability to explore the real Pareto front (PF) without gradient information [19]. The key of the multi-objective evolutionary algorithm (MOEA) is that the algorithm achieves a balance between diversity and convergence, so as to effectively generate a set of Pareto optimal solutions that satisfy multiple conflicting objectives at the same time.

The multi-objective particle swarm optimization (MOPSO) [20,21,22], multi-objective ant colony optimization (MOACO) [23], and multi-objective bee colony optimization (MOBCO) [24,25] are typical multi-objective evolutionary algorithms that are widely used in practical engineering problems. Among them, MOPSO has the advantages of simple implementation and fast convergence speed, which is widely used in the optimization of practical engineering problems [26,27,28]. However, due to the existence of non-dominated solutions, the quality of the solution cannot be easily assessed, which brings difficulties to the selection of guiding particles. In particular, as the dimension of decision variables gradually increases, the MOPSO can no longer explore effectively.

The current research focuses on how to improve the optimization efficiency on the basis of balancing the convergence and diversity of the algorithm. At present, the improvement for MOPSO is mainly divided into three aspects. On the one hand, the multi-objective problem is solved using a competitive strategy. The competitive particle swarm optimization algorithm introduces a competitive mechanism, which selects the appropriate evolution direction through the game between particles [29,30,31]. On the other hand, multi-objective problems are optimized through a decomposition strategy. Yu et al. proposed a decomposition-based comprehensive learning particle swarm optimization algorithm, which optimizes each objective in the early stage and cooperates with each other in the later stage [32]. In addition, some scholars improve the guided particle selection strategy to optimize multi-objective problems. Zhu et al. proposed a novel archive-guided velocity update method to guide the transfer of the population [33]. Li et al. introduced a strategy based on ranking dominance to select the elite particles, which can obtain a more diverse and uniform distribution external archive [34]. The above algorithms introduce strategies on MOPSO to improve the performance of the algorithm, but as the target dimension increases, a single strategy will make the algorithm enter into long-term ineffective exploration, or even fall into local optimum. The IAMOPSO algorithm proposed in this paper focuses not only on convergence and diversity, but also on the efficiency of exploration.

To evaluate the effect of reconstruction, not only the accuracy, but also the robustness (the ability to resist external interference when the sensor is installed) should be considered. In this paper, the reconstruction accuracy is evaluated using the root mean square error (RMSE) of the actual displacement and theoretical displacement, and the robustness is evaluated by the composite sensitivity matrix.

This paper is organized as follows. Section 2 briefly introduces the inverse finite element method for beam elements, and then establishes an optimization model of sensor distribution based on reconstruction accuracy and stability metrics. Section 3 proposes an improved adaptive multi-objective particle swarm optimization (IAMOPSO) algorithm. Section 4 applies IAMOPSO to the sensor layout problem on the deformation reconstruction of the circular beam model. The numerical and experimental verification results confirm that IAMOPSO is an excellent tool for sensor placement in iFEM.

## 2. Review of Beam Deformation Reconstruction through iFEM

### 2.1. Inverse Finite Element Method

The inverse finite element method (iFEM) is described in detail in Ref. [11]. The principle is briefly described in this section. As shown in Figure 1, for a Timoshenko beam, the deformation of the neutral axis can be represented by the displacement along each axis and the rotation angle around each axis. The displacement of the neutral axis u can be expressed as Equation (1).
(1)$$u={\left[u,v,w,\theta_{x},\theta_{y},\theta_{z}\right]}^{Τ}$$

The displacement of any point on the beam cross-section along the direction *x*, direction *y*, and direction *z* can be represented by the neutral axis displacement:(2)$$\begin{array}{l} u_{x}(x,y,z)=u(x)+z\theta_{y}(x)-y\theta_{z}(x)\\ u_{y}(x,y,z)=v(x)-z\theta_{x}(x)\\ u_{z}(x,y,z)=w(x)+y\theta_{x}(x) \end{array}$$

Define arbitrary section strain as: $e(u)={\left[e_{1},e_{2},e_{3},e_{4},e_{5},e_{6}\right]}^{Τ}$. The strain along the axis at any point is the derivation of Equation (2) with respect to x:(3)$$\begin{array}{l} \varepsilon_{x}(x,y,z)=e_{1}(x)+ze_{2}(x)+ye_{3}(x)\\ \gamma_{\mathrm{xz}}(x,y)=e_{4}(x)+ye_{6}(x)\\ \gamma_{\mathrm{xy}}(x,y)=e_{5}(x)-ze_{6}(x) \end{array}$$

The section strain can be described as Equation (4).
(4)$$\begin{array}{c} e_{1}(x)=u_{x}(x),e_{2}(x)=\theta_{y,x}(x),e_{3}(x)=-\theta_{z,x}(x)\\ e_{4}(x)=w_{x}(x)+\theta_{y}(x),e_{5}(x)=v_{x}(x)-\theta_{z}(x),e_{6}(x)=\theta_{x,x}(x) \end{array}$$

Based on the least square method, iFEM minimizes the theoretical strain and experimental strain errors. The fitting function ϕ(u) between the theoretical section strain e(u) and the actual section strain $e^{\varepsilon }$ is defined as follows:(5)$$\phi (u)={\left\|e(u)-e^{\varepsilon }\right\|}^{2}$$

The displacement at any section position can be represented by the shape function N(x) and the nodal degrees of freedom $u_{e}$:(6)$$u(x)=N(x)u_{e}$$

After that, the strain at any section position can be calculated by Equation (7).
(7)$$e(u)=B(x)u_{e}$$
where B(x) is called the strain function matrix, which is the derivative of N(x). In order to minimize the error of theoretical strain and actual strain, let the derivative function of Equation (5) with respect to u be zero, then Equation (8) can be obtained.
(8)$$k^{e}u_{e}=f^{\varepsilon }$$

$k^{e}$ and $f^{\varepsilon }$ can be expressed as:(9)$$\begin{array}{l} k^{e}=\sum_{k=1}^{6}w_{k}k_{k}^{e}\;\;\;k_{k}^{e}=\frac{L}{n}\sum_{i=1}^{n}\left[B_{k}^{T}(x_{i})B_{k}(x_{i})\right]\\ f^{\varepsilon }=\sum_{k=1}^{6}w_{k}f_{k}^{e}\;\;\;f_{k}^{e}=\frac{L}{n}\sum_{i=1}^{n}\left[B_{k}^{T}(x_{i})e_{k}^{\varepsilon }(x_{i})\right] \end{array}$$
where L denotes the element length, n is the number of section strain distributions on the neutral axis, and $x_{i}$ represents the location where the section strains are evaluated; $w_{k}$ is a weighting factor considering the mutual influence of the axial stretching, bending, twisting, and transverse shearing; $e_{k}^{\varepsilon }(x_{i})$ are the section strains at the location $x_{i}$, which is computed from strain-sensor values $\varepsilon_{2}^{*}$, shown in Ref. [11]. $k^{e}$ is a function of $x_{i}$, whereas $f^{\varepsilon }$ depends on $x_{i}$ and the measured strain values $\varepsilon_{2}^{*}$.

After splicing multiple inverse finite elements, Equation (10) can be obtained.
(10)KU=F
where K is global stiffness matrix, which is related to section position $x_{i}$; F is the global load vector, which depends on $x_{i}$ and $\varepsilon_{2}^{*}$. The matrix-vector equation provides a solution for the nodal freedom of structural deformation, $U=K^{-1}F$. Then, through Equation (6), the deformation of any section position can be obtained by interpolation. According to Equation (2), the deformation of any point can be calculated eventually.

### 2.2. Indexes to Evaluate the Effect of Deformation Reconstruction

After reconstructing the displacement field of the structure based on the iFEM, this section proposes two indicators to evaluate the effect of the reconstruction: reconstruction accuracy and robustness metrics. The former estimates the accuracy of reconstruction, and the latter reflects the stability of the reconstructed model.

When the sensor layout scheme X is selected, the reconstruction accuracy index RRMSE(X) is defined as follows:(11)$$RRMSE(X)=\frac{RMSE(X)}{max(disp^{F})}\times 100\%$$
(12)$$RMSE(X)=\sqrt{\frac{1}{N}\sum_{i=1}^{N}{\left(disp_{i}^{F}(X)-disp_{i}^{P}(X)\right)}^{2}}$$
where $disp^{F}$ represents the theoretical displacement, which refers to the deformation value from simulation; $disp^{P}$ represents the displacement obtained by deformation reconstruction; *N* is the number of check points.

To explicitly the influence of strain–sensor values on reconstruction stability, Equation (10) can be transformed as
(13)$$U=A(X)\varepsilon_{2}^{*}$$
where A(X) is referred as a strain-deformation converted matrix, which is only related to the location of sensors; $\varepsilon_{2}^{*}$ is the measurement strain vector, and U is the nodal freedom of structural deformation.

The method proposed in Ref. [35] is used to measure the reconstruction stability, as shown in Equation (14).
(14)$$\cos(\theta_{i}(X))=\frac{A_{i}(X)P(X)A_{i}^{T}(X)}{\left\|A_{i}(X)\right\|\times \left\|P(X)A_{i}^{T}(X)\right\|}$$
(15)$$P_{i}(X)={\bar{A}}_{i}^{T}(X){\left({\bar{A}}_{i}(X){\bar{A}}_{i}^{T}(X)\right)}^{-1}{\bar{A}}_{i}(X)$$
where $A_{i}(X)$ is the i-th row vector of matrix A(X), and ${\bar{A}}_{i}(X)$ consists of the remaining row vectors of A(X) minus $A_{i}(X)$. $\theta_{i}(X)$ is calculated for each row vector, and the robustness index selects its maximum value. The reconstruction robustness index is defined as
(16)$$RBI(X)=max(\theta_{i}(X))\;i=(1,2,\ldots m)$$
where m denotes the dimension of the structural node degrees of freedom. The bigger the value of RBI(X), the higher the error tolerance of matrix A(X) to sensor errors.

The purpose of the sensor layout problem in iFEM is to find a sensor scheme that achieves a balance between reconstruction accuracy and reconstruction stability. The optimization model of sensor distribution is described as follows:(17)$$\begin{array}{l} Minmize\;\mathrm{Index}(X)=[RRMSE(X),-RBI(X)]\\ X=(X_{1},X_{2},\ldots ,X_{6}) \end{array}$$
where the number of sensors used in the reconstruction method is six; Index(X) represents objective function, consists of accuracy index RRMSE(X) and robustness index −RBI(X). The IAMOPSO proposed in Section 3 is used to find sensor schemes that minimize both metrics simultaneously.

## 3. Optimization Algorithm

### 3.1. Multi-Objective Particle Swarm Optimization Algorithm

The Particle Swarm Optimization (PSO) algorithm is a swarm intelligence optimization algorithm, which was proposed by Kennedy and Eberhart [36]. It guides individuals to transfer through the coordination among individuals, as shown in Equation (18) [37].
(18)$$X_{i}(t+1)=X_{i}(t)+V_{i}(t+1)$$
(19)$$V_{i}(t+1)=w\cdot V_{i}(t)+c_{1}\cdot r_{1}\cdot (pb_{i}(t)-X_{i}(t))+c_{2}\cdot r_{2}\cdot (gb_{i}(t)-X_{i}(t))$$
where $pb_{i}(t)$ is the individual optimal, which represents the optimal position of particle $X_{i}(t)$ in the past generation. $gb_{i}(t)$ is the group optimal, which represents the optimal location explored by all individuals in the group. w is the inertia weight (all w in Section 3 represent inertia weight). $c_{1}$ and $c_{2}$ are acceleration factors. $r_{1}$ and $r_{2}$ are random values between 0 and 1. The velocity at the next moment $V_{i}(t+1)$ of the particle $X_{i}(t)$ is determined by the current velocity $V_{i}(t)$, individual optimal $pb_{i}(t)$, and group optimal $gb_{i}(t)$. The particle moves from $X_{i}(t)$ to $X_{i}(t+1)$ with an updated velocity $V_{i}(t+1)$, as shown in Figure 2.

In multi-objective problems (MOP), there may be multiple exclusive objective functions, which make the quality of the solutions impossible to be simply compared. The Pareto dominance relation is defined as follows: If for X and Y in the decision space,$\forall j\in \left\{1,2,\ldots m\right\},f_{j}(X)\leq f_{j}(Y)$,and $\exists j\in \left\{1,2,\ldots m\right\},f_{j}(X)<f_{j}(Y)$, m represents the number of objective functions, then the decision variable X is defined to dominate the decision variable Y. The final solution of MOP is not a single solution, but a set of non-dominated solutions.

Throughout the exploration, for MOP, the following issues need to be considered:How to guide particles to converge:

Algorithm convergence is the foundation of an optimization algorithm. As shown in Figure 2, the global optimum $gb_{i}$ and the individual optimum $pb_{i}$ influence the iterative direction in each generation; hence, it is crucial to adopt an appropriate guided particle selection strategy.

2.How to balance the exploration and exploitation of algorithm:

If the algorithm focuses on exploitation and ignores exploration, particles may converge rapidly and fall into a local optimum. If the algorithm focuses on exploration and neglects exploitation, the algorithm may explore ineffectively.

3.How to search the target space fully:

For problems with complex shapes of the Pareto front, the algorithm needs to fully explore the target space to ensure the integrity of the Pareto front.

4.How to improve exploration efficiency in a high-dimensional space:

Algorithms need to explore the Pareto front in a finite time, so the particles should explore as efficiently as possible. In the guidance process, the particle information needs to be fully utilized to help the particles make better transfer strategies.

### 3.2. Improved Adaptive Multi-Objective Particle Swarm Optimization Algorithm

In this section, in order to balance the convergence and development of the algorithm and improve the efficiency of the algorithm, the IAMOPSO introduces the following four strategies on the basis of MOPSO. Moreover, the strategies employ external candidate solution (ECS) set consisting of non-dominated solutions obtained by the dominance relationship, which is updated after each particle transfer.

#### 3.2.1. Initialization Strategy

MOPSO usually adopts the method of uniform initialization to generate the initial population. When the variable dimension is high and the target space is large, the population needs to explore for a long time. In addition, there are few particles in the ECS set in the early stage, which tends to trap the population in local optima. To solve this problem, an initialization method based on a single-objective problem is proposed in this paper.

Before formally entering the multi-objective optimization with *m* objectives, *m* times of single-objective optimization are carried out, respectively. In the single-objective optimization, the population size generally takes $N_{\mathrm{Ini}}<N/5$ and the number of iterations takes $T_{\mathrm{Ini}}\;<\;T/10$, where N is the number of particles and T is the maximum number of iterations. Some particles of each generation retained in the single-objective optimization process, which constitute the initial population ${P=P}_{\mathrm{IniF}1}\cup P_{\mathrm{IniF}2}\cdots \cup P_{\mathrm{IniFm}}$, where $P_{\mathrm{Ini}-\mathrm{Fi}}\;$ represents the population composed of particles retained by single-objective optimization for the *i*-th objective. Take *m* = 2 as an example, Figure 3 shows the composition of the initial population for dual-objective optimization. The red particles are the particles retained in the process of optimizing the objective function F1, while the blue particles are the particles retained in the process of optimizing the objective function, F2. This is because, in the whole optimization process of single-objective optimization, each particle tends to cluster around the optimal position of the objective. Using the initialization strategy proposed in this section to synthesize a multi-objective initial population can maintain this trend and produce initial solutions with better performance.

#### 3.2.2. Adaptive Inertia Weight Strategy

The adaptive inertia weight strategy makes the algorithm focus on exploration ability in the early stage and exploitation ability in the later stage. It can be observed from the Equation (19) that the algorithm has better exploration ability when w>1, and gradually converges when w<1. IAMOPSO introduces a time factor to the inertia weight w, and maps the Sigmoid function to the curve of w.
(20)$$w(t)=w_{\max}-(w_{\max}-w_{\min})\cdot sigmoid(\frac{\Delta S\times t}{T}-S_{\max})$$
(21)$$sigmoid(x)=\frac{1}{1+\exp(-x)}$$
(22)$$\Delta S=S_{\max}-S_{\min}$$
where *T* represents the maximum number of iterations.

Taking different $S_{\min}$ and $S_{\max}$ can dynamically adjust w, so as to control the exploration and exploitation of the algorithm. As shown in Figure 4, $w_{\max}$ takes 1.2 and $w_{\min}$ takes 0.8. The three curves in the figure use ($S_{\min}$, $S_{\max}$) as (−6, 6), (−4, 6), (−6, 4), which map out different w and control the transition of the algorithm.

#### 3.2.3. Guided Particle Selection Strategy

In order to improve the efficiency of IAMOPSO, this paper adopts the following adaptive method to select the guiding particle: When ECS is improved, the population maintains the state of exploitation, and selects the particle closest to the origin in ECS set as gb. when the ECS is not improved in a long time, the population enters the state of exploration, and selects an individual in the sparse region of the ECS set as gb. The ECS set is partitioned by angle in each generation, as shown in Figure 5. When the population is in an exploitation state, the algorithm calculates the Euclidean distance between each point and the origin. The distance between particle A and the origin is the smallest, so particle A is selected as gb. When the population is in exploration state, particle A can no longer effectively guide particles, while particle B in the sparse region can guide particles to explore a uniformly distributed Pareto front. Particle B is selected as gb in this iteration.

Since different objective functions have different scales in practical applications, it is necessary to normalize the values of each objective function before partitioning. The maximum and minimum values of particles in the ECS set are used for normalization, as in Equation (23). $F_{j}(X_{i})$ is the actual target value, and $F_{j}^{\prime }(X_{i})$ is the value after normalization of the particle $X_{i}$. As shown in Figure 5, the particle M corresponds to $F_{1}^{\min}$ and $F_{2}^{\max}$, while the particle N corresponds to $F_{1}^{\max}$ and $F_{2}^{\min}$. In this way, the particles in the ECS set are normalized into a unit hypercube of [0,1], which facilitates the estimation of the evolutionary state.
(23)$$F_{j}^{\prime }(X_{i})=(F_{j}(X_{i})-F_{j}^{\min})/(F_{j}^{max}-F_{j}^{\min})\;F_{j}^{max}=\max_{x_{i}\in P_{\mathrm{ECS}}}F_{j}(X_{i})\;F_{j}^{\min}=\min_{x_{i}\in P_{\mathrm{ECS}}}F_{j}(X_{i})$$

In the early stage of the iterative algorithm, there are few solutions in the ECS. If there are many partitions, many continuous sparse regions will be generated, resulting in low efficiency. In the later stage of the iterative algorithm, the number of solutions in the ECS set is large. If there are too few partitions, the sparse regions cannot be effectively identified during partitioning. In order to solve the above problems, the algorithm adopts an adaptive partition strategy according to the number of particles in the ECS set, which guides the population to explore sparse regions more efficiently.

#### 3.2.4. External Candidate Solution Set Maintenance Strategy

In the later stage of the iterative algorithm, the number of particles in ECS set explodes, resulting in too much computational time. To reduce the selection pressure, some redundant particles need to be dynamically removed when the number of particles in ECS set reaches the threshold. The maintenance of non-dominated solutions focuses on how to reduce the selection pressure while maintaining the shape of PF.

Here, we use the partitioning strategy proposed in Section 3.2.3. We specify the maximum capacity of the ECS set $N_{MaxA\mathrm{rc}}$. After each iteration, the ECS set is updated according to the Pareto dominance relationship. If the number of particles in the ECS set $N_{A\mathrm{rc}}>N_{MaxArc}$, the algorithm maintains the external candidate solution. The algorithm first calculates the number of redundant particles, and then divides the ECS set into D regions: $A=\left\{A_{1,},A_{2},\cdots ,A_{D}\right\}.$ Select a set of dense regions $A_{dense}$, and randomly delete particles in dense region $A_{dense}$ according to the proportion.

A dense region is defined as: if the number of particles in the region k satisfy $\rho (A_{k})-\rho (A_{sparse})\;\geq \;N_{remove}$, where Asparse=ρAimin(Ai),i∈[1,D], then $A_{k}$ belongs to $A_{dense}$, where $\rho (A_{i})$ is the particle density (number of particles) in region $A_{i}$.

As shown in Figure 6, the maximum capacity of ECS $N_{MaxArc}$ is set to 20, while the current number of particles in ECS $N_{A\mathrm{rc}}$ is 25. At this time, redundant particles need to be dynamically removed, and the number of redundant particles $N_{remove}$ is 5. Calculating the particle density in each region $\rho (A_{i})$, it can be found  Asparse=ρAimin(Ai)=A5, and $\rho (A_{sparse})=\rho (A_{5})=1$. Since $\;\rho (A_{1})=9$ and $\;\rho (A_{3})=6$, which satisfy $\rho (A_{k})-\rho (A_{sparse})\geq N_{remove}$, then $A_{1}$ and $A_{3}$ belong to the dense region. Three redundant particles in $A_{1}$ and two redundant particles in $A_{3}$ are randomly removed according to the proportion (the red particles in Figure 6 will be dynamically deleted).

#### 3.2.5. Algorithm Framework

The general framework of IAMOPSO is described as follows (see Algorithm 1).
**Algorithm 1:** Framework of IAMOPSO**Input:** Initialized population size NIni, initialized number of iterations TIni, number of particles N, maximum number of iterations T, maximum capacity of ECS set NMaxArc, adaptive inertia weight parameters wmax, wmin, Smin, and Smax, and number of initial partitions D = 3**Termination condition**: The maximum number of iterations is reached.**Step 1: Initialize**1. Use the initialization strategy to generate the initial population.2. Select the non-dominated solutions in the initial population and store them in the ECS set.**Step 2: Iteration**1. The inertia weight w is adaptive.2. Estimate the evolutionary state. At the beginning of the iteration, the population is in the exploitation state.3. Partition the target space. Calculate the population density in each partition.4. Select gb according to the evolution state.5. Calculate V and perform particle transfer.6. Update ECS set to ensure that all non-dominated solutions explored are stored in the ECS set.7. If the number of external candidate solutions exceeds the limit, maintain ECS set and remove redundant particles.8. Determine whether the termination condition is reached, if it is satisfied, output the ECS set, otherwise enter the next iteration.**Output**: ECS set

### 3.3. Technical Route

The sensor layout optimization employing the proposed IAMOPSO algorithm comprises three steps, which are illustrated in the schematic diagram of Figure 7. After the model of sensor distribution is established, the accuracy index RRMSE(X) and robustness index RBI(X) of sensor scheme are used as fitness functions in the IAMOPSO algorithm to evaluate the quality of each scheme. After optimization with IAMOPSO, a set of non-dominated sensor schemes are obtained. Then, the evaluation of IAMOPSO is achieved through numerical and experimental evaluation on circular beams and complex models.

## 4. Algorithm Evaluation

### 4.1. Numerical Examples

In this section, to evaluate the effect of the IAMOPSO algorithm on the sensor layout in the iFEM, a cantilever beam is reconsidered for the simulation example compared with the MOPSO proposed in Ref. [18]. The model employs a beam of homogeneous solid with the engineering simulation finite element analysis, which is divided into 200 elements and the cross-section of the beam is divided into 36 segments, as shown in Figure 8. The length of the beam model is L=660 mm, the outer radius is R=13 mm, and the inner radius is r=11.5 mm. The target space is a series of position points, which carry strain information. The Young’s modulus is E=7.03 GPa, the Poisson’s ratio is v=0.3, and the density is $p=2700\;\mathrm{kg}/m^{3}$.

Different loads are subjected to the free end of cantilever beam, which are formed by the simultaneous application of two loads, as shown in Table 1.

Six sensors (a sensor distribution scheme) were used to reconstruct the displacement field of the beam, and 200 checkpoints were selected to calculate the RRMSE. Based on the optimization model of sensor distribution proposed in Section 2, the IAMOPSO algorithm is used to optimize the layout of sensor, while the MOPSO proposed in Ref. [17] is used to compare and verify the performance of IAMOPSO. All sensor distribution schemes obtained and the Pareto front with IAMOPSO are shown in Figure 9.

As shown in Figure 9b, the distribution of the Pareto front shows that the reconstruction effect with IAMOPSO exceeds with MOPSO significantly. Moreover, by observing the particle transfer in the optimization process, it can be found that the MOPSO lacks the ability to jump out of the local optimum, which is inefficient. Meanwhile IAMOPSO can make full use of the information of particles and jump out of the local optimum, which is efficient.

The typical positions C1 and C2 on the Pareto front are selected for comparison with the position C3 obtained by the MOPSO algorithm, and the position information is shown in Table 2. In order to intuitively measure the reconstruction effect, the maximum error (${\mathrm{ERR}}^{\max}$) is introduced, which is defined as Equation (24).
(24)$${\mathrm{ERR}}^{\max}=\max\left|disp_{i}^{F}(X)-disp_{i}^{P}(X)\right|,i=1,\ldots ,200$$
where $disp_{i}^{F}(X)$ represents the theoretical displacement of *i*-th check points and $disp_{i}^{P}(X)$ represents the displacement of the *i*-th check point obtained by deformation reconstruction.

Table 3 compares the ${\mathrm{ERR}}^{\max}$ of the three typical schemes without interference, where Max_disp represents the maximum theoretical deformation displacement. The result shows that the reconstruction accuracy of the schemes C1 and C2 are better than that of the scheme C3.

In order to verify the robustness, interferences are applied to three schemes, as shown in Equation (25). When the sensor distribution scheme is determined, these interferences randomly distributed in the error range are added 500 times on each sensor independently, and the maximum error $RRMSE^{\mathrm{err}}$ is recorded.
(25)$$\Delta x_{i}\in \left(-0.3,0.3\right),\Delta y_{i}\in \left(-0.05,0.05\right),\Delta z_{i}\in \left(-0.05,0.05\right),i=1,...6$$

Table 4 shows the values of robustness and accuracy index on three schemes. The RRMSE of the three schemes is 2.6%, 2.8% and 3.4% without interference, while they degrade to 6.9%, 5.1% and 11.3% with interference applied, respectively. Obviously, C2 has excellent resistance to interference, which is more suitable for complex working conditions.

In conclusion, according to the above analysis results, the sensor scheme C2 dominates C3 on all evaluation metrics, which indicates that the optimization effect of IAMOPSO obviously exceeds that of MOPSO.

### 4.2. Numerical and Experimental Validation

In this section, to demonstrate the universality of IAMOPSO algorithm for sensor layout applications, numerical and experimental verifications are performed on a complex plate-beam structure, which consists of three beams and one plate. The length of the beam is $L_{c}=2200\;\mathrm{mm}$, the outer radius is R=20 mm, and the inner radius is r=16 mm. The length of the board is 1500 mm, the width is 400 mm, and the thickness is 8 mm. The board is 470 mm from the fixed end. The distance between the main beam (middle beam) and the two symmetrical sub-beams is 130 mm. The ribs are 400 mm long, 50 mm wide, and 20 mm thick. The distances from the six ribs to the fixed end are 250 mm, 520 mm, 870 mm, 1220 mm, 1570 mm, and 1920 mm, respectively, as shown in Figure 10. The Young’s modulus is E=7.03 GPa, the Poisson’s ratio is v=0.3, and the density is $p=2700\;\mathrm{kg}/m^{3}$. In the simulation, a static load of 350 N was applied to the free end of the beam.

Considering robustness and accuracy as indexes, the sensor distribution model is established. Then, the IAMOPSO algorithm is used to optimize the sensor placement, and the Pareto front is shown in Figure 11. Non-dominated schemes **P_1_**~**P_4_** on PF are selected for detailed description, whose location information is shown in Table 5.

Figure 12 shows the corresponding sensor location of the four schemes **P_1_**~**P_4_** in the model. It can be observed that a position of scheme **P_2_** is relatively close to the rib, hence scheme **P_2_** is not given priority considering the difficulty of installation. The remaining layout schemes are finally selected according to the actual engineering needs.

In the experiments, two types of sensors are used, one is the fiber Bragg grating (FBG) sensor that captures the surface strain, and the other is the position sensor that captures the true displacement of the marked points (check points). According to the sensor distribution scheme obtained by simulation, six fiber FBG sensors are installed on the main beam to reconstruct the displacement field, and position sensors are installed at the marked points of the plate, the positions of which are captured by a 3D optical measurement device (NDI Optotrak Certus, Northern Digital Incorporation, Waterloo, ON, Canada), as shown in Figure 13.

Remove the unreasonable sensor scheme **P_2_**, and select the sensor schemes **P_1_** and **P_3_** with higher accuracy for experimental tests with different end node loads in the Z direction. The accuracy analysis is shown in Table 6, and ${\mathrm{iFEM}}^{\max}$ represents the maximum reconstructed deformation, while ${\mathrm{NDI}}^{\max}$ represents maximum experimental deformation measured by NDI.

It can be observed from Table 6 that under the load of 200 N, the maximum experimental deformation ${\mathrm{NDI}}^{\max}$ is 100.29 mm. With scheme **P_1_**, the maximum reconstructed displacement ${\mathrm{iFEM}}^{\max}$ is 103.09 mm, while with scheme **P_3_**, ${\mathrm{iFEM}}^{\max}$ is 104.26 mm. The error is controlled within 4 mm. Under the load of 350 N, the ${\mathrm{NDI}}^{\max}$ is 144.09 mm, the ${\mathrm{iFEM}}^{\max}$ with scheme **P****_1_** is 148.21 mm, while ${\mathrm{iFEM}}^{\max}$ with scheme **P_3_** is 149.21 mm, the error is controlled within 5 mm, and the RRMSE can also be kept within 5%. The results demonstrate that the reconstruction effect of the sensor schemes **P_1_** and **P_3_** is excellent.

Since the reconstruction accuracy of the sensor layout scheme **P_1_** is similar to that of the sensor layout scheme **P_3_**, the scheme **P_1_** is selected for visualization. Figure 14 shows the deformation measured by NDI and reconstructed deformation with scheme **P_1_** under two concentrated loading conditions, and the reconstructed deformation is coincided with the reference value from the NDI, which indicates that the reconstruction effect is excellent with scheme **P_1_**.

In conclusion, IAMOPSO can well solve the sensor distribution problem in iFEM. In practical engineering applications, under complex working conditions, on the basis of meeting the accuracy requirements, a sensor distribution scheme with higher stability should be selected, while under other conditions, a sensor distribution with high accuracy has priority.

## 5. Conclusions

This paper focuses on an optimization algorithm that can be better applied to sensor layout problems. This paper first constructs a multi-objective optimization model of sensor distribution in iFEM based on the accuracy and robustness metrics. To balance the optimization of the two objectives and improve the efficiency of the algorithm, this paper proposes an improved adaptive multi-objective particle swarm optimization algorithm, which introduces an initialization strategy, adaptive inertia weight strategy, guided particle selection strategy, and ECS set maintenance strategy. The results on the beam model validate the performance of the algorithm. Finally, the algorithm is applied to the deformation reconstruction of the complex plate-beam model, which can maintain high reconstruction accuracy and stability. The numerical and experimental results demonstrate that the optimization algorithm IAMOPSO proposed in this paper can well solve the sensor distribution problem in iFEM. Future work will focus on solving high-dimensional optimization problems in complex structural reconstruction.

## Figures and Tables

**Figure 1 sensors-22-05203-f001:**
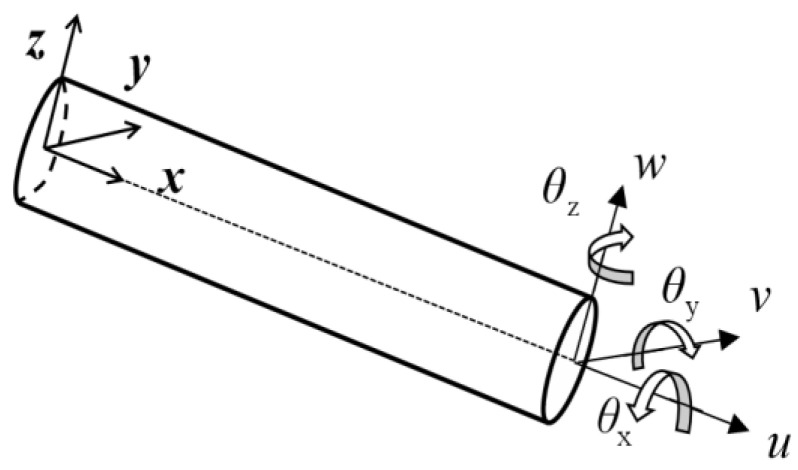
Schematic diagram of Timoshenko beam.

**Figure 2 sensors-22-05203-f002:**
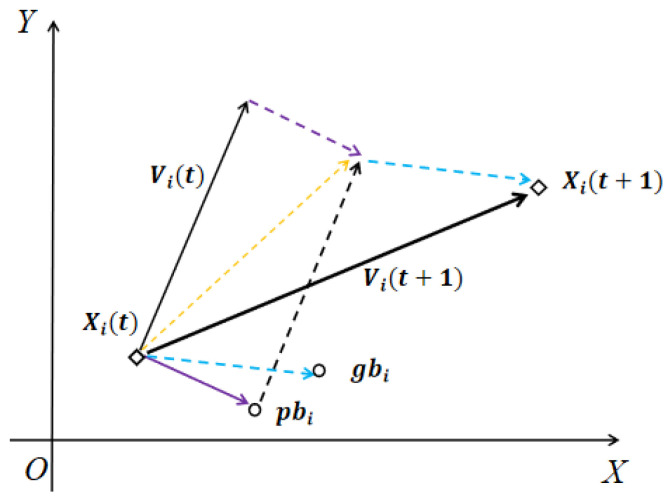
The transfer process of particle swarm optimization.

**Figure 3 sensors-22-05203-f003:**
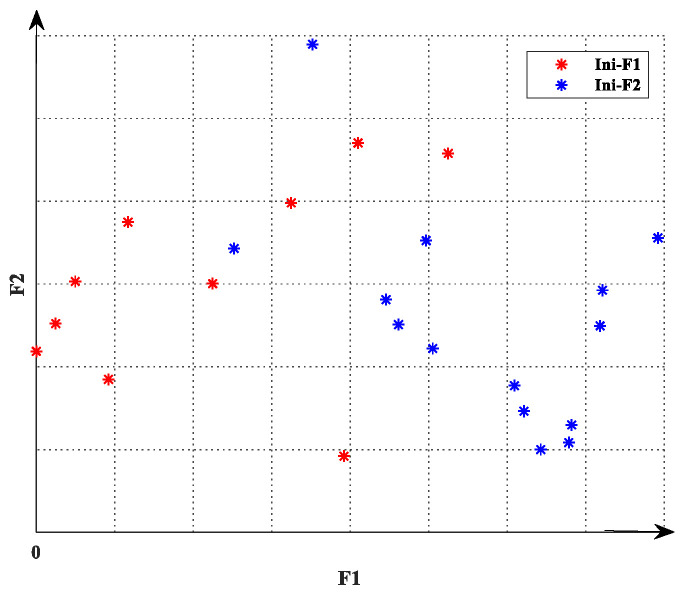
The composition of the initial population in dual-objective optimization.

**Figure 4 sensors-22-05203-f004:**
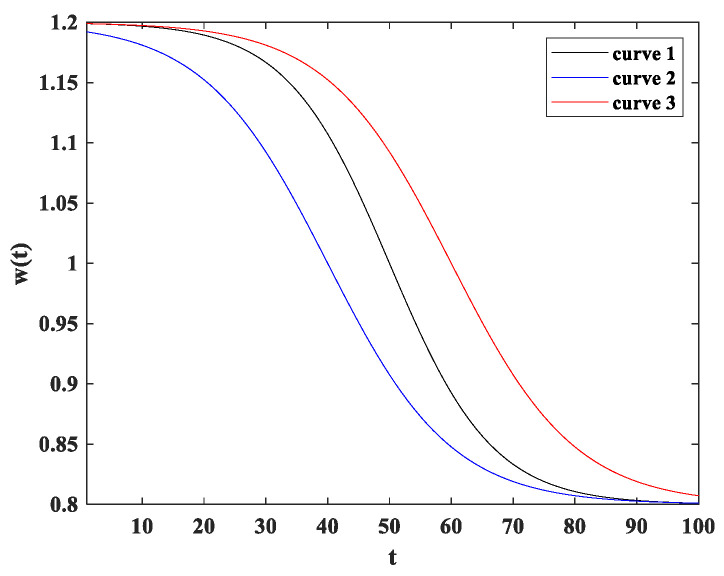
Adaptive process of inertial weight.

**Figure 5 sensors-22-05203-f005:**
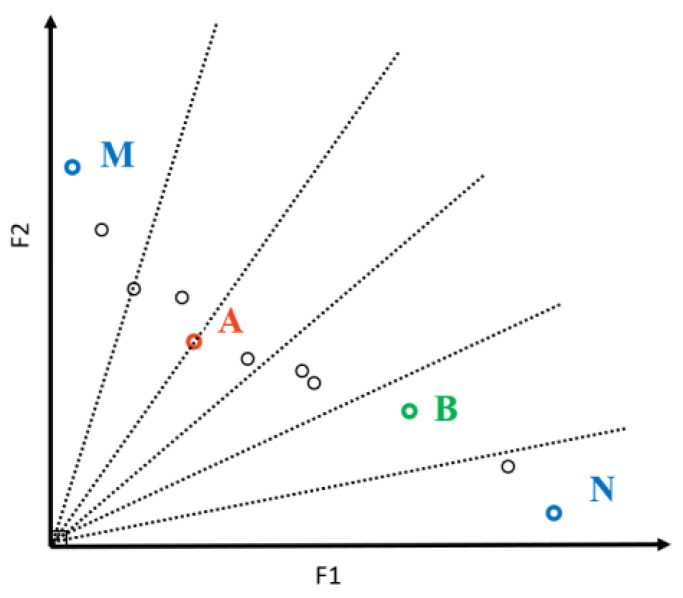
Globally optimal particles in different evolution states.

**Figure 6 sensors-22-05203-f006:**
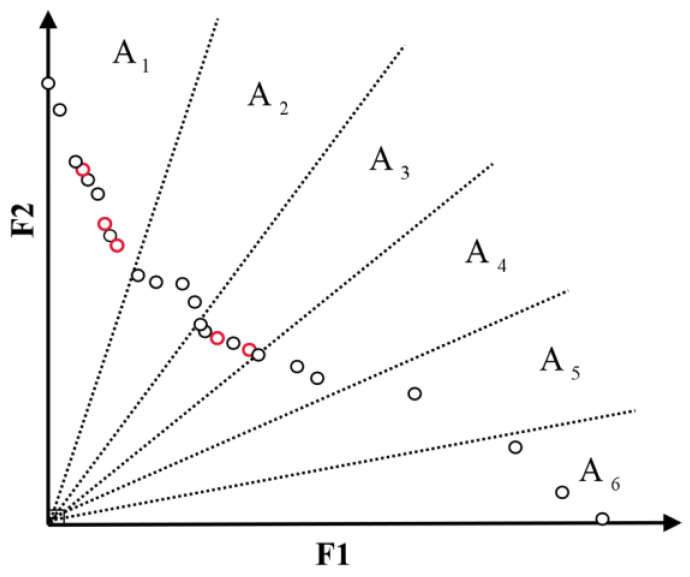
Maintenance of external candidate solution set.

**Figure 7 sensors-22-05203-f007:**
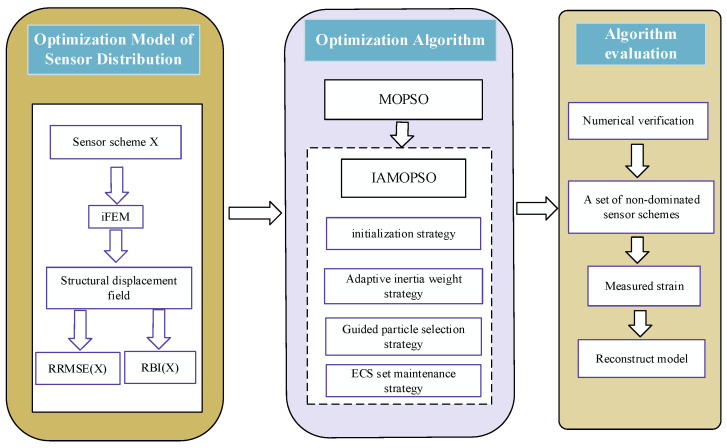
Technical route for sensor layout optimization employing IAMOPSO.

**Figure 8 sensors-22-05203-f008:**
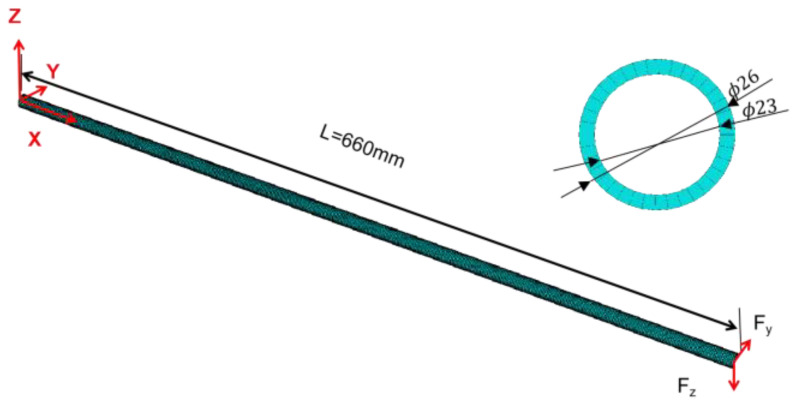
Finite element model of the beam.

**Figure 9 sensors-22-05203-f009:**
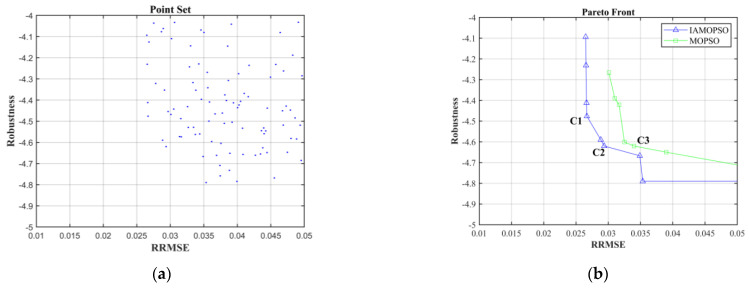
Optimization results: (**a**) The objective function corresponding to the sensor distribution scheme explored in the IAMOPSO optimization process; (**b**) comparison of Pareto front explored by IAMPSO and MOPSO.

**Figure 10 sensors-22-05203-f010:**
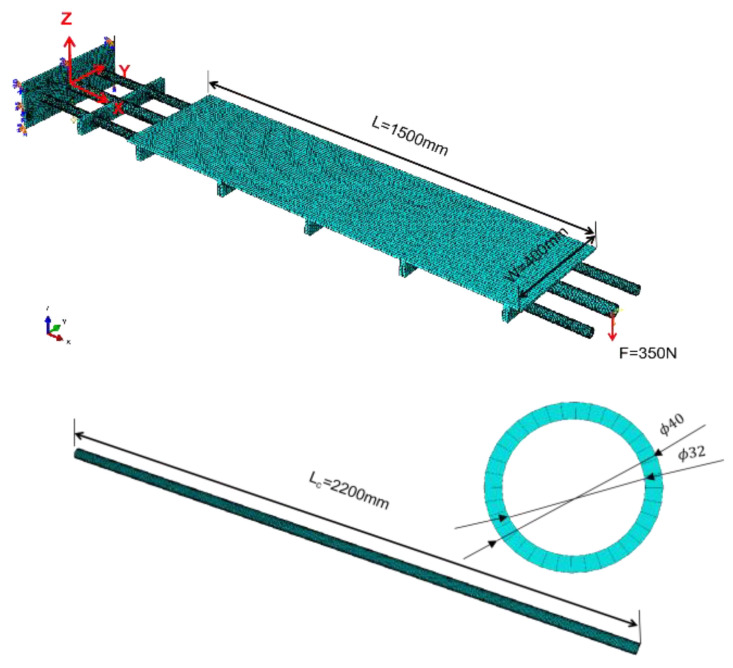
Complex plate–beam structure under concentrated load.

**Figure 11 sensors-22-05203-f011:**
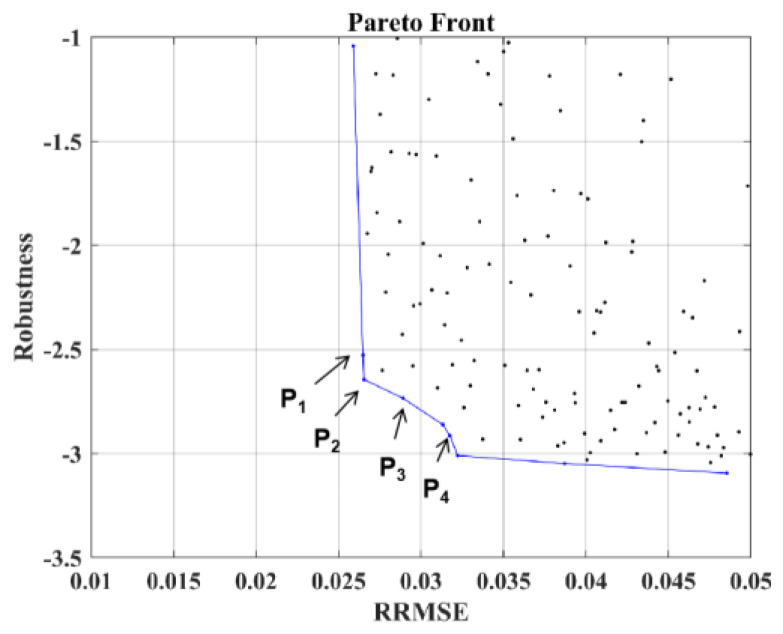
Pareto front based on IAMOPSO.

**Figure 12 sensors-22-05203-f012:**
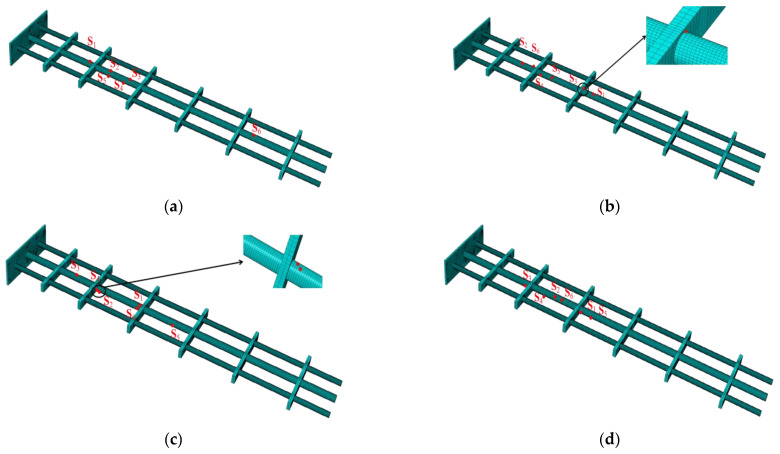
The position corresponding to **P_1_** (**a**), **P_2_** (**b**), **P_3_** (**c**), and **P_4_** (**d**) in the model.

**Figure 13 sensors-22-05203-f013:**
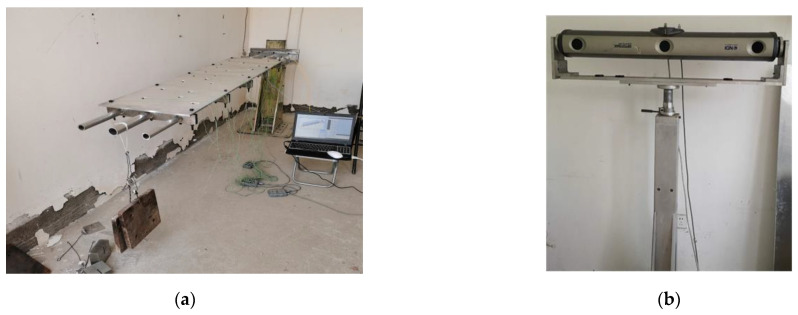
Experimental platform: (**a**) Loading on end-node of entire structure; (**b**) NDI Optotrak Certus.

**Figure 14 sensors-22-05203-f014:**
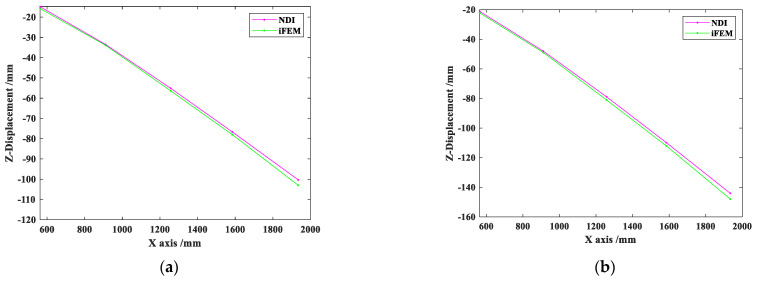
Displacement in Z direction with **P_1_** under concentrated load of 200 N (**a**) and 350 N (**b**).

**Table 1 sensors-22-05203-t001:** Load applied at the end of the beam.

Load	Heavy Load (N)	Moderate Load (N)
$F_{y}$	250	120
$F_{z}$	200	100

**Table 2 sensors-22-05203-t002:** The distribution scheme of the sensors with IAMOPSO (C1, C2) and MOPSO (C3) (mm).

	C1	C2	C3
$\varepsilon_{1}$	(190.00, −10.32, −10.00)	(40.00, −10.32, −10.00)	(217.00, −10.32, 10.00)
$\varepsilon_{2}$	(257.00, 13.00, 0.00)	(168.00, −12.69, 3.47)	(343.00, −11.79, 6.84)
$\varepsilon_{3}$	(182.00, −3.47, 12.69)	(318.00, 3.47, −12.69)	(212.00, −8.33, 10.86)
$\varepsilon_{4}$	(408.00, −8.33, −10.86)	(288.00, 0.00, 13.00)	(328.00, −10.32, 10.00)
$\varepsilon_{5}$	(123.00, 11.79, −6.84)	(38.00, −12.69, 3.47)	(252.00, −12.69, −3.47)
$\varepsilon_{6}$	(348.00, −3.47, 12.69)	(248.00, 10.86, 8.33)	(248.00, 6.84, −11.79)

**Table 3 sensors-22-05203-t003:** Comparison of ${\mathrm{ERR}}^{\max}$ under three sensor distribution schemes (mm).

Load (N)	Direction	Max_Disp	C1	C2	C3
Heavy Load	X	0	$6.2\times 10^{-4}$	$6.8\times 10^{-4}$	$7.9\times 10^{-4}$
	Y	39.2	0.07	0.10	0.19
	Z	31.5	0.05	0.06	0.08
Middle Load	X	0	$5.5\times 10^{-4}$	$6.3\times 10^{-4}$	$5.8\times 10^{-4}$
	Y	18.9	0.06	0.08	0.11
	Z	16.3	0.06	0.07	0.09

**Table 4 sensors-22-05203-t004:** Comparison of reconstruction accuracy and robustness metrics.

Optimizer	Scheme	RRMSE	Robustness Index	$RRMSE^{\mathrm{err}}$
IAMOPSO	C1	0.026	−4.41	0.069
	C2	0.028	−4.62	0.051
MOPSO	C3	0.034	−4.61	0.113

**Table 5 sensors-22-05203-t005:** Sensor location information of scheme **P_1_** to **P_4_** (mm).

	P_1_	P_2_	P_3_	P_4_
$\varepsilon_{1}$	(485.00, 6.84, −18.79)	(955.00, −6.84, −18.79)	(840.00, 19.69, −3.47)	(890.00, 15.32, 12.85)
$\varepsilon_{2}$	(640.00, 6.84, −18.79)	(415.00, −0.00, −20.00)	(550.00, 17.32, 10.00)	(695.00, 10.00, −17.32)
$\varepsilon_{3}$	(785.00, 3.47, −19.69)	(880.00, −10.00, −17.32)	(380.00, 3.47, −19.69)	(475.00, 3.47, 19.69)
$\varepsilon_{4}$	(745.00, 10.00, 17.32)	(565.00, 18.79, −6.84)	(545.00, 18.79, 6.84)	(620.00, 17.32, 10.00)
$\varepsilon_{5}$	(625.00, 19.69, 3.47)	(645.00, 6.84, −18.79)	(1080.00, 19.69, 3.47)	(965.00, 15.32,1 2.85)
$\varepsilon_{6}$	(1660.00, −20.00, 0.00)	(495.00, −10.00, −17.32)	(835.00, −17.32, 10.00)	(750.00, 10.00, −17.32)

**Table 6 sensors-22-05203-t006:** Analysis of reconstruction accuracy under two concentrated loads.

Index	200 N	350 N
${\mathrm{NDI}}^{\max}$	−100.29 mm	−144.09 mm
${\mathrm{iFEM}}^{\mathrm{P1\_max}}$	−103.09 mm	−148.21 mm
${\mathrm{iFEM}}^{\mathrm{P3\_max}}$	−104.26 mm	−149.03 mm
$RRMSE^{P1}$	0.037	0.039
$RRMSE^{P3}$	0.042	0.047

## Data Availability

Not applicable.

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
