# Peer review of "Improved Adaptive Multi-Objective Particle Swarm Optimization of Sensor Layout for Shape Sensing with Inverse Finite Element Method"

_sensors, 2022, doi:10.3390/s22145203_

Round 1

Reviewer 1 Report

All comments and remarks are given in the attached pdf file.

Author Response

Dear reviewer:

       Firstly, thank you for your careful review of our manuscript and give us the meaningful advice to improve the quality of the manuscript.

       Our responses to your comments are given in the attached file.

With best regards!

Reviewer 2 Report

The manuscript presented the configuration of sensor layout for inverse finite element using an improved adaptive multi-objective particle swarm optimization. The work will be reconsidered after the following questions and comments are replied.

(1)    The introduction missed many representative references of the topic. Some related works are:

-An inverse finite element method for beam shape sensing: theoretical framework and experimental validation

-Efficient Bayesian sensor placement algorithm for structural identification: a general approach for multi‐type sensory systems

-An area coverage and energy consumption optimization approach based on improved adaptive particle swarm optimization for directional sensor networks

(2)    Regarding the reconstruction robustness index, why the matrix A is defined as the inverse of the K matrix?

(3)    Why the index given in Eq. (13) can quantify the robustness. According to Ref. [32], the quantity shown in Eq. (13) indicates another physical meaning.

(4)    In Eq. (15), what do X and F(X) refer?

(5)    According to the optimization problem shown in Eq. (15), why RRMSE(X) and -theta_i(X) are comparable in the minimization? RRMSE is in percentage and theta is an angle.

(6)    For the presented improved adaptive multi-objective particle swarm optimization (IAMOPSO) algorithm, what are the new ideas from other existing IAMOPSO?

(7)    The authors claimed that the presented strategies can “balance the convergence and development of the algorithm and improve the efficiency of the algorithm“. It is unclear what are the contributions of different strategies and how to design the subjective settings.

(8)    In Section 3.2.2, why the sigmoid function is applied?

(9)    How to assign the “maximum number of iterations”? How to guarantee it is a proper choice?

(10)In Table 4, why the robustness of the three schemes is different?

(11)In the theoretical development and numerical examples, how the measurement noise of the sensors is taken into account?

(12)In the demonstrations, a comparison with other sensor layouts should be presented and discussed.

(13)The English of the manuscript should be carefully polished.

Author Response

(The authors gave the same response as above.)

Reviewer 3 Report

An improved multi-objective particle swarm algorithm is proposed to rationally optimize the sensor layout. However, I feel the manuscript need to be improved significantly. The comments are given below:

1. As the focus of this work, the influence of strain sensor distribution on the reconstruction effect should be pointed out at the beginning of the Introduction, and lines 25-51 should be streamlined.

2. The overview of Optimization of sensor layout in iFEM and MOPSO in the Introduction is fragmented, which makes it difficult for the reader to find the need for this work. It is suggested that the authors reorganize this section.

3. We cannot visualize the proposed initialization strategy from Figure 3. Moreover, in lines 216-219, the expression Because in the single-objective optimization, each particle tends to gather near the optimal point of the objective. Retaining this trend to synthesize the initial population of multi-objectives can produce initial solutions with better performance is confusing. I believe that in both single-objective and multi-objective optimization, the initial population generation is random unless a specific mechanism is used. There is no such thing as each particle tends to gather near the optimal point of the objective. Obviously, the manuscript does not show this specific mechanism in detail.

4. Section 4, Algorithm evaluation, presents a comparative analysis of the reconstruction effect of IAMOPSO and MOPSO. However, only a large amount of data results are described. The authors should explore in depth how the improvement strategies of the proposed algorithm work to increase the persuasiveness of the results and the depth of the manuscript.

5. In lines 58 to 60 and 75 to 76 of the article, there is a contradiction in the description of the performance of the MOPSO algorithm, and MOPSO does not perform better than other optimization algorithms for some problems compared to other evolutionary optimization algorithms. For example, MOEA, NSGA-III.

6. In the article, the authors state that Optimization of sensor layout in iFEM is a multi-objective optimization problem where model gradient information is not easy to obtain, but it is not clear why the problem is a multi-objective problem? That is, whether there is a conflict between the reconfiguration accuracy objective and the robustness objective. If there is no conflict, a single-objective optimization model can be considered to solve this sensor layout optimization problem.

7. What is described in lines 182 to 198 is more like what the PSO algorithm should consider rather than what MOPSO should consider.

8. In the article, what do N and T in line 214 refer to? Why should be divide by 5 and 10, respectively?

9. Figure 3 is intended to illustrate what problem in the initialization process? Where F1 and F2 refer to what, respectively?

10. In the article, why is the sigmoid function used when making adaptive changes to the weights? What is its motivation and theoretical basis? How are the parameters determined?

11. The language of the paper could be further streamlined and improved.

Author Response

(The authors gave the same response as above.)

Reviewer 4 Report

·         The novelty of this work needs to be elaborated more. Why this study is so important? How it differentiates from the other methods e.g. Ref. [16].

·         Line 58:

The authors stated that “However, due to the lack of a strategy to jump out of the local optimum, MOPSO is prone to ineffective exploration, with poor effect and low efficiency” , this sentence needs more evidences and arguments.

·         Equation 15 is depending on which references, if any, or fundamentals have been employed?

·         It is necessary to provide references in equations 9,16,18, 22.

·         Have you considered how the finite element size affects the accuracy of the results? What is the size of the used element?

·         The paper lacks details in regards to the setup of the simulation.

·         It would be useful for increasing both the visibility and the size of Figure 10.

·         It would be useful to offer further information on the experimental procedure.

·         The research work is well conducted, but the article in the present form does not provide an advance in current knowledge. At least, I cannot see it.

Author Response

(The authors gave the same response as above.)

Round 2

Reviewer 2 Report

The authors replied the raised questions and suggestions. 

Reviewer 3 Report

This manuscript has been greatly improved and has good academic value. It is recommended for publication.

Reviewer 4 Report

The authors have satisfactorily addressed most of my concerns and made several improvements to the article.